# Microbiota in Autism Spectrum Disorder: A Systematic Review

**DOI:** 10.3390/ijms242316660

**Published:** 2023-11-23

**Authors:** Zuzanna Lewandowska-Pietruszka, Magdalena Figlerowicz, Katarzyna Mazur-Melewska

**Affiliations:** Department of Infectious Diseases and Child Neurology, Poznan University of Medical Sciences, 60-572 Poznan, Poland; 69511@student.ump.edu.pl (Z.L.-P.); mfiglerowicz@ump.edu.pl (M.F.)

**Keywords:** autism spectrum disorder, microbiota, functional gastrointestinal disorder, behavior, systematic review

## Abstract

Autism spectrum disorder (ASD) is a complex neurodevelopmental condition characterized by several core symptoms: restricted interests, communication difficulties, and impaired social interactions. Many ASD children experience gastrointestinal functional disorders, impacting their well-being. Emerging evidence suggests that a gut microbiota imbalance may exacerbate core and gastrointestinal symptoms. Our review assesses the gut microbiota in children with ASD and interventions targeting microbiota modulation. The analysis of forty-four studies (meta-analyses, reviews, original research) reveals insights into the gut microbiota–ASD relationship. While specific microbiota alterations are mixed, some trends emerge. ASD children exhibit increased Firmicutes (36–81%) and Pseudomonadota (78%) and decreased Bacteroidetes (56%). The Bacteroidetes to Firmicutes ratio tends to be lower (56%) compared to children without ASD, which correlates with behavioral and gastrointestinal abnormalities. Probiotics, particularly *Lactobacillus*, *Bifidobacterium*, and *Streptococcus* strains, show promise in alleviating behavioral and gastrointestinal symptoms (66%). Microbiota transfer therapy (MTT) seems to have lasting benefits for the microbiota and symptoms in one longitudinal study. Prebiotics can potentially help with gastrointestinal and behavioral issues, needing further research for conclusive efficacy due to different interventions being used. This review highlights the gut microbiota–ASD interplay, offering potential therapeutic avenues for the gut–brain axis. However, study heterogeneity, small sample sizes, and methodological variations emphasize the need for comprehensive, standardized research. Future investigations may unveil complex mechanisms linking the gut microbiota to ASD, ultimately enhancing the quality of life for affected individuals.

## 1. Introduction

Autism spectrum disorder (ASD) is a neurodevelopmental disorder with a complex and not fully understood etiology. The core symptoms include stereotypical, limited interests and behavior, as well as difficulties in communication and social functioning [1]. Children with autism experience difficulties in daily life, including in basic care for themselves and their surroundings, motor skills, and behavior. These symptoms can be misunderstood or unaccepted by their family or peers [2]. Patients frequently report, also through their parents, somatic symptoms that have a significant impact on their overall well-being and quality of life [3]. Some of the most frequent ones are gastrointestinal functional disorders, which are reported in up to 70% of children with ASD [4].

There is increasing evidence of an imbalance in the gut microbiome in patients with ASD. These anomalies have the potential to exacerbate the fundamental, behavioral, and gastrointestinal manifestations of autism [5]. The microbiome is widely acknowledged as a crucial component of the so-called “gut–brain axis”, a term that denotes the reciprocal connection between the central nervous system and the gastrointestinal (GI) system, including the enteric nervous system [3]. For centuries, the gut–brain axis has been an area of interest regarding the link between mental and physical symptoms and the condition of the GI system [6], and the newest research supports these observations. The microbiota, for instance, can regulate hormonal or inflammatory responses, which impact digestive tract function as well as brain development and maturation [7]. Several neuropsychiatric disorders, such as Alzheimer’s disease, schizophrenia, anxiety, and depression, have been observed to be exacerbated by the microbiota. In children with ASD, research on the changes in microbiota composition in comparison to neurotypical individuals and potential therapy focused on concomitant factors lowering quality of life has been a trend in the field in the last few years [5,8,9,10,11]. Because studies have presented observations from different populations and utilized several interventions, the overall conclusions remain equivocal.

In this systematic review, the authors aimed to present the current state of knowledge about the microbiota composition in children with autism spectrum disorder, as well as the interventional possibilities that were observed to be effective in single studies in these patients.

## 2. Methods

We performed a systematic literature review according to the Preferred Reporting Items for Systematic Reviews and Meta-Analyses (PRISMA) Checklist [12] to analyze the current state of knowledge on gut microbiota abnormalities in children with autism spectrum disorder. We searched the PubMed Central database using the following search query based on Medical Subject Headings (MeSH): ((((((((autism) OR ASD) OR autistic) AND microbiome) OR microbiota) AND microflora) NOT murine) NOT mice) NOT rat. We obtained 10,983 results.

The studies were screened for inclusion by the following eligibility criteria: (1) the study was a systematic review, a meta-analysis, or an original study (randomized controlled trial, prospective and retrospective cohort studies, case–cohort studies, open-label study, or observational study); (2) the subjects of the study were children (aged 0–18 years); (3) the paper concerned the microbiota composition or an intervention involving the microbiota; (4) the researchers provided information about the age structure of the groups and the method used for the microbiota assessment; (5) the article was written in English; (6) the full text was accessible online. The excluded studies comprised (1) animal models of autism; (2) microbiota analyses in other disorders; (3) papers such as non-systematic reviews, case reports, and others with methods marked as not rigorous enough.

The Rayyan Tool [13] was used for initial screening, which resulted in the removal of duplicates of ineligible articles, and a single withdrawn article. Further, we excluded manuscripts that did not meet the eligibility criteria. The authors screened the remaining articles and selected 44 papers to be included in this systematic review. The selected studies comprised six meta-analyses (MAs) [14,15,16,17,18,19], nine systematic reviews (SRs) [20,21,22,23,24,25,26,27,28], and 29 original manuscripts: five randomized controlled studies (RCTs) [5,8,9,10,11] and 24 non-randomized controlled studies (non-RCTs) [29,30,31,32,33,34,35,36,37,38,39,40,41,42,43,44,45,46,47,48,49,50,51,52]. The data from the analyzed original papers came from 1123 ASD patients and 816 healthy controls in total. The selection process is visualized in the PRISMA chart [12] in Figure 1.

The quality of the studies included in this systematic review was based on the assessments performed individually by two reviewers. The quality of the meta-analysis and systematic review was assessed using the A MeaSurement Tool to Assess Systematic Reviews (AMSTAR) 2 Checklist [53]. The results are presented in Figure 2. Non-randomized controlled trials were evaluated with the Risk of Bias in Non-Randomized Studies of Exposure and of Interventions [54]. The results are included in Figure 3 and Figure 4, respectively. Randomized controlled trials (RCTs) were assessed with the Revised Cochrane Risk of Bias Tool for Randomized Trials [55]. The risk of bias was marked as high if any domain was rated as high, and it was marked as low if no domains were rated as high. The assessment is visualized in Figure 5. In the case of uncertainty in the assessment, the final score was given after obtaining a consensus between the reviewers. A detailed description of the included papers can be found in Table 1 for non-interventional original studies on the gut microbiota, Table 2 for non-interventional original studies on the oral microbiota, Table 3 for interventional original studies on the gut microbiota, and Table 4 for meta-analyses and systematic reviews.

## 3. Results and Discussion

### 3.1. Differences in Microbiome between ASD and Non-ASD Patients

The data on the composition of the gut microbiota are equivocal. The details are presented in the Table 5. There was a lack of a discernible trend observed in multiple papers regarding the abundance of the genera in ASD patients in comparison to neurotypical patients. Nonetheless, there were a limited number of genera for which iterative trends could be observed. Moreover, in some cases, the difference between ASD patients and controls was observed only for a limited group of patients—for example, those being on a restrictive diet—which led to some of the data being excluded from the analysis of general trends.

In the Actinobacteria phylum, the genera observed to be less abundant were *Bifidobacterium* (67% of papers analyzing this genus—eight non-RCTs [29,30,31,34,36,41,48,50] and two RCTs [9,11] vs. 27%—four non-RCTs [32,38,45,52]) and *Eggerthella* (67%—two non-RCTs [31,32]).

In Bacteroidetes, the overall abundance of the phylum was lower in ASD patients; however, the trend was not strong (56%—five non-RCTs [29,30,36,44,50] vs. 33%—two non-RCTs [31,52] and one RCT [11]). On the genus level, the lower abundance was observed primarily in *Alistipes* (100%—three non-RCTs [30,43,49]), while a weak trend for higher abundance was observed in *Bacteroides* (50%—one RCT [8] and four non-RCTs [31,46,48,52], with the last one observed only in *B. coprococcus* and *B. vulgaris* and one elevated only in *B.fragilis* vs. 50%—five non-RCTs [29,30,36,49,52], with the last one observed only in *B. fragilis*).

In Firmicutes, the overall abundance was higher, but the trend was not strong (81%—one RCT [8] and eight non-RCTs [33,36,37,38,44,49,50,52] five family-specific (36% and 45% family-specific) vs. seven non-RCTs (63%) [30,32,33,37,44,50,52], four family-specific (27% and 36% family-specific). The genera with higher abundance included *Blautia* (60%—three non-RCTs [49,50,52], vs. 40%—two non-RCTs [29,36]), *Clostridium* (57%—four non-RCTs [31,45,46,52] vs. 29% three non-RCTs [30,48,49]—one family-specific (14%)), *Dorea* (100%—two non-RCTs [43,49]), *Lactobacillus* (80%—four non-RCTs [33,38,43,46] vs. 20%—one non-RCT [29]), *Oscillospira* (100%—one RCT [11], one non-RCT [31]), and *Ruminococcus* (67%—one RCT [11], three non-RCTs [30,31,38] vs. 33%—two non-RCTs [29,36]), while lower abundance included *Dialister* (83%—five non-RCTs [29,30,43,50,52] vs. 17%—one non-RCT [49]), *Phascolarctobacterium* (67%—two non-RCTs [30,49] vs. 33%—one non-RCT [29]), *Streptococcus* (71%—five non-RCTs [31,32,33,34,35,36,37,38,39,50,51] vs. 14%—one non-RCT [52]), *Roseburia* (57%—four non-RCTs [30,31,36,38] vs. 29%—two non-RCTs [29,52]), and *Veilonella* (75%—three non-RCTs [43,50,51] vs. 25%—one non-RCT [29]).

In Pseudomonadota, the overall abundance was higher in ASD patients (78%—seven non-RCTs [30,31,33,36,37,46,50] vs. 22%—two non-RCTs [32,51]). On the genus level, *Klebsiella* (100%—two non-RCTs [38,50]) and *Sutterella* (100%—two non-RCTs [29,30]) abundance was elevated, while that of *Escherichia* (75%—three non-RCTs [50,51,52] vs. 25%—one non-RCT [48]) and *Shigella* (100%—two non-RCTs [49,50]) was lowered.

In Verrucomicrobia, *Akkermansia* abundance was lower in ASD patients (86%—six non-RCTs [29,30,48,50,51,52] vs. 14%—one RCT in non-elimination diet patients only [8]).

The ratio of Bacteroidetes to Firmicutes was more often observed to be lower (56%—five non-RCTs [38,44,49,50,52] vs. 44%—one RCT [11], three non-RCTs [31,43,51]).

In MAs and SRs, the abundance was higher in ASD patients in Pseudomonadota on the phylum level [15,23,27] and on the genus level: *Parabacteroides* [15,23,27], *Clostridium* [14,15,23,27], *Dorea* [23,27], *Ruminococcus* [18,23,27], *Faecalibacterium* [15,19,23], and *Sutterella* [14,23,27]. On the other hand, the abundance was lowered in *Bifidobacterium* [14,15,18,23,27], *Bacteroides* [17,18,23], *Prevotella* [14,20,23,27], *Enterococcus* [18,23], *Streptococcus* [14,20,23,27], *Fusobacterium* [23,27], *Escherichia* [18,20], and *Akkermansia* [20,27]. The details are presented in the Table 6.

Only three non-RCTs concerned the oral microbiota; thus, the data are scarce. The details are presented in the Table 7. In one of the papers, lowered bacterial diversity was observed, similar to the gut microbiota. The iterative results included higher abundance of *Rothia* (67%) [39,40] (although one of the papers observed such a trend only in dental samples), *Streptococcus* (67%) [39,47], and Proteobacteria (67%) (specifically *Haemophilus*) [39,40], and lower abundance of Firmicutes (67%) [39,40] and *Fusobacteriota* (67%) [39,40]. Higher abundance of *Moryella* correlated with better verbal skills and a lower occurrence of early autism symptoms. *Ralstonia* positively correlated with social skills. There was a positive correlation between verbal, non-verbal, and total intelligence quotients and *Tannerella* abundance, while both *Ralstonia* and *Weeksellaceae* correlated negatively with these quotients. *Saccharibacteria* abundance correlated with poorer communication and stereotypical behavior, while a larger *Tannerella* amount was related to better communication [40].

Although the microbiota composition had a low correlation with the occurrence of social difficulties, it correlated clearly with the core symptoms and the severity of GI symptoms. The core symptoms were more intense in patients with a larger amount of *Bacteroides*, *Clostridium*, and *Desulfovibrio*, and with a lower Bacteroidetes/Firmicutes ratio [45].

The patients with GI symptoms had a larger amount of some genera in the Firmicutes phylum (especially *Dialister*, *Fusicatenibacter*, and *Turicibacter*) and a smaller amount of *Phascolarctobacterium* [49]. Moreover, the occurrence of such symptoms correlated with the abundance of *Asteroleplasma*, *Thalassospira*, *Burkholderia*, *Comamonadaceae*, *Fusobacteriales*, *Prevotellaceae*, *Mobilincus*, *Clostridium* (especially *cluster XVIII*), and *Desulfovibrio*, and both the *Escherichia/Shigella* and Bacteroidetes/Firmicutes ratio [42]. Constipation occurrence was negatively correlated with *Gemmiger* and *Ruminococcus* abundance [43].

Importantly, the elimination diet led to significant differences in the microbiota composition. Patients on an elimination diet presented with lower abundance of *Bifidobacterium* and higher abundance of Bacteroidetes and Firmicutes on the phylum level and *Faecalibacterium* and *Roseburia* on the genus level [8]. Patients with food selectivity had a greater amount of *Prevotella*, *Anaerophilum*, *Clostridium*, and *Salmonella*, alongside a higher *Escherichia/Shigella* ratio [46].

In two of the meta-analyses, the authors attempted to select a microbiota composition that could be used as a biomarker for ASD. On the genus level, *Prevotella*, *Roseburia*, *Ruminococcus*, *Megasphaera*, and *Catenibacterium* were marked out. The strains with biomarker potential included *Prevotella buccae*, *Bifidobacterium longum*, *Streptococcus thermophilus*, *Enterobacter cloacae*, *Klebsiella oxytoca*, *Eubacterium hallii*, *Clostridium ramosum*, *Erysipelotrichaceae bacterium 6_1_45*, *Eubacterium siraeum*, and *Lautropia mirabilis* [17,25].

When the siblings of the ASD children were compared with independent controls, they had a lower Bacteroidetes/Firmicutes ratio as well [45].

### 3.2. Probiotic Interventions

A probiotic based on different strains of *Lactobacillus*, *Bifidobacterium*, and *Streptococcus* (LBS probiotic) was used in two RCTs [5,10] and one nRCT [45]. The authors of one RCT [11] and one nRCT [41] treated the patients with a probiotic with strains of *Lactobacillus* and *Bifidobacterium* (LB) only, while one nRCT applied a non-specific (ns) probiotic therapy [36]. Apart from these, five SRs analyzed the influence of the probiotics on core and associated symptoms. In one of these, therapy based only on different *Lactobacillus* strains (L) was used [26]. The overall data analysis included in the SRs indicated that probiotics have a positive influence on the functioning of patients with ASD.

LBS probiotics seemed to improve various aspects of difficult behavior, including irritability, social avoidance, stereotypies, hyperactivity, and non-compliance (33% of reports) [45]. LB probiotics based on *L. acidophilus*, *L. rhamnosus*, and *B. longum* also led to a significant decrease in the severity of difficulties with communication, sensory and cognitive awareness, and sociability (50%) [41]. However, LB therapy using *L. rhamnosus*, *B. animalis*, and *B. lactis* led to the worsening of the behavior (50%) [21]. The nRCT using non-specific probiotics and the data in SRs supported these observations [26]. The correlation was especially observed for behavioral abnormalities, especially irritability, but less so communication difficulties. Among L therapy, *L. plantarum* did not improve behavior, while *L. acidophilus* had a positive influence on concentration [21].

The LBS (33%) and ns probiotic therapy seemed to also improve sleep quality and, for ns therapy, eating pattern abnormalities as well. The stress perceived by the parents was reduced. The anxiety in children with ASD was reduced after LBS probiotic therapy; however, the improvement was not statistically significant, while mood was improved in the research with ns probiotic therapy [5,26]. Overall behavioral efficacy was observed in 67% of reports of LBS therapy and 50% of those of LB therapy.

The SRs showed mixed results for probiotics in general regarding functional gastrointestinal symptoms (FGID) [16,21,22,23,24,26], while singular papers indicated some effectiveness. FGID were reduced after LBS probiotic therapy. Statistical importance was observed in one of the two RCTs [5,10] (67%). An improvement after LB therapy was observed in one RCT [11] (50%). It was supported by the data analyzed in one of the systematic reviews [23]. It was also observed in one nRCT after ns therapy. In the group of children with and without FGID, there was no improvement in the core symptoms after such probiotic therapy. However, in the group comprising only children without FGID, such an improvement was visible, while children with FGID showed an improvement in adaptive functioning in such aspects as receptive, domestic, and coping skills or sensory profile [36]. Therapy based only on *L. plantarum* did not seem to improve FGID [26]. Overall FGID efficacy was observed in 67% of reports of LBS therapy and 50% of those of LB therapy.

LBS therapy led to an increase in the Bacteroidetes/Firmicutes ratio. Moreover, it lowered the *Desulfovibrio* level and *Lactobacillus* absolute amount, while the relative amount of the latter increased [45]. Probiotics seemed to improve *Bifidobacterium* and *Enterococcus* abundance [41] and lower that of *Desulfovibrio* and *Clostridium* [45].

### 3.3. Microbiota Transfer Therapy Interventions

Microbiota transfer therapy (MTT) was used in one longitudinal nRCT published in two papers. The authors observed an increase in the abundance of *Bifidobacterium*, *Prevotella*, and *Desulfovibrio*. The effect on the former two lasted at least for two years when a follow-up assessment was performed. In some patients, MTT also increased the abundance of *Roseburia*, *Ruminococcus*, and *Faecalibacterium*, and the general abundance was higher after the transfer. The patients after MTT also presented an improvement both in FGID and in behavior. These results were also observed two years after the intervention, and the improvement in FGID led to the more visible alleviation of behavioral issues, suggesting that FGID can potentiate difficult behavior [34,35]. The data are supported by the results of systematic reviews [23,26].

### 3.4. Prebiotic Therapy

Prebiotics were less commonly used as an intervention in the analyzed research. In general, prebiotics seemed to lower alpha-diversity, with a simultaneous increase in *Bifidobacteriales* abundance and a decrease in Clostridium abundance.

One RCT focused on the usage of a prebiotic galactosigosaccharide mixture (B-GOS) in both a group with an unrestricted diet and a group with dietary restrictions. The prebiotic led to significant differences in the microbiota composition in the unrestricted diet group, such as an increase in *Bifidobacterium* spp., *Ruminococcus* spp., *Lachnospiraceae*, *Eubacterium dolchum*, *Saccharibacteria*, and *Mogibacteriaceae*, while the restricted diet group presented with a predominance of *B. longum* in the *Bifidobacterium* spp. group. The intervention led to a non-statistically important improvement in FGID (while the children on the restrictive diet presented with less FGID at the baseline). Approximately one quarter of children presented with some improvement in sleep, and the children on the restrictive diet presented with an improvement in social behavior [8].

Another RCT compared the combination of a prebiotic (bovine colostrum product) and Bifidobacterium probiotic treatment with a treatment based only on the prebiotic. The authors suggested the existence of four different baseline enterotypes: *Prevotella* based, *Bifidobacterium* based, *Bacteroides* based, and mixed. The intervention led to no change in enterotypes. However, the prebiotic-only intervention led to an improvement in GI symptoms and behavior in such aspects as irritability, stereotypy, hyperactivity, and lethargy. The combination of pre- and probiotics provided all the patients with some improvement in their GI symptoms; however, the prebiotic only was reported by parents to show greater efficacy. The combined treatment also led to improved behavior but only in the lethargy aspect [9].

The combination of both pre- and probiotics, analyzed in two SRs, seemed to improve both behavior (three RCTs, one non-RCT) and GI symptoms (one RCT, one non-RCT) [16,26]. The overall efficacy of prebiotics, due to the diverse products used in the analyses, is difficult to assess.

### 3.5. Other Interventions

In one of the systematic reviews, a paper on the role of vitamin A supplementation was included. According to the authors, such therapy led to an increase in the Bacteroidetes/Bacteroides and Bacteroidetes/Firmicutes ratios; however, the intervention had no influence on the ASD core symptoms [26].

Vancomycin therapy seemed to have a positive effect on behavior; however, the results were not visible after the antibiotic was reduced, and the strength of the evidence was too weak to form a clear conclusion [23].

### 3.6. Discussion

In this systematic review, we analyzed 16 papers on the gut microbiota composition, three papers on the oral microbiota composition, 10 studies on different interventions, and 15 meta-analyses or systematic reviews. Although the current understanding of the composition of the microbiota in patients with ASD appears to be in a state of development, there are some promising findings that have been uncovered through the research conducted to date.

The GI symptoms seem to be highly correlated with the abundance of the bacteria from the phyla Firmicutes and Bacteroidetes. Importantly, Firmicutes is more abundant in patients with ASD, both in comparison to neurotypical patients and in relation to Bacteroidetes abundance, which is lowered. The amount of both Firmicutes and Bacteroidetes is increased in those individuals who are on the exclusion diet. In the context of ASD patients, the exclusion diet relates to the elimination of casein and gluten. It is frequently followed by children with ASD, varying in reports from 13% to 88% of families [56,57]. There are no clear data on the beneficial effects of such diets, apart from in those with proven food allergies and celiac disease. Positive results were observed in some groups of patients; however, the trials were performed in small groups [58] or based on parent-report questionnaires [59] and results therefore might be subjective. The other studies showed little to no positive influence on both the core and GI symptoms [57]. The exclusion diet led to an increase in the abundance of *Faecalibacterium* and *Prevotella*, which were observed to be correlated with irritable bowel syndrome symptoms in neurotypical patients [60,61].

Food selectivity and other feeding disturbances are a separate, but common, phenomenon in ASD children, observed in 45–89% of patients. Low fiber intake, along with high content of simple carbohydrates and saturated fats [62], can lead to an important shift in the gut microbiota. This type of diet was observed to relate to the FGID symptoms. However, contrary to the results observed in the ASD patients, it correlated with decreased *Prevotella* abundance [63]. Interestingly, the prevalence of *Prevotella* might relate to the so-called “westernization” of the diet, also connected with the higher consumption of simple carbohydrates and unsaturated fats [4], which corresponds to a typical selective diet in ASD patients. On the other hand, *Escherichia* and *Salmonella*, the bacteria more abundant in ASD patients with food selectivity, produce hydrogen sulfide, which is detrimental to the gut immune response and its mobility and secretion, along with visceral hypersensitivity [64]. This makes it challenging to determine whether the FGID symptoms connected with food selectivity are caused by a high-carbohydrate diet or whether the shift in microbiota plays an important role here. Interestingly, a high-saturated-fat diet leads to a decrease in *Bifidobacterium* abundance [65], which was also observed in ASD patients on the exclusion diet; however, it also lowers the abundance of *Bacteroides* and *Clostridium*, contrary to the observations in ASD children. From other genera, *Clostridiales*, elevated in ASD patients especially with FGID symptoms, were observed to correlate with proinflammatory cytokines in the gut mucosa [66]. Moreover, Firmicutes, especially *Ruminococcaeae*, were found to be more abundant in patients with non-celiac gluten sensitivity [67], while *Fusobacterium* and *Desulfovibrio* were linked to FGID symptoms in two other studies [68,69]. The severity of the core symptoms correlated with higher *Bacteroides* and *Desulfovibrio* levels, observed also in the context of FGID. A connection between FGID and the core or behavioral difficulties in ASD was previously reported [70], and the correlation on the microbial level might support this observation. However, a comparison of the results of the studies included in this systematic review showed no clear difference between the western diet and non-western diet groups.

The study of potential microbial biomarkers was focused mainly on Firmicutes (*Roseburia*, *Ruminococcus*, *Megasphaera*, and *Catenibacterium*, and some species of *Clostridium*, *Eubacterium*, *Erysipelatrichaceae*, and *Streptococcus*) and Bacteroidetes (*Prevotella*, and some species of *Klebsiella* and *Lautropia*), as well as some species of Bifidobacterium (*Actinobacteria*) and Enterobacter (*Pseudomonadota*). Importantly, the Bacteroidetes/Firmicutes ratio seems to be not only a predictor of the core and GI symptoms but also of the typical microbiota of ASD children.

The discrepancies in microbiota abundance in different papers can be associated with a variety of factors influencing the gut microbiota, such as age, sex, diet, comorbidities and chronic drugs, previous antibiotic therapy, or method of delivery. With a more thorough examination, the paradoxically elevated abundance of Bacteroidetes in some papers is observed in two non-RCTs, where the control group consisted only or partially of siblings, which could have influenced the results. The only study with different results was Zou et al. For the Bacteroidetes/Firmicutes ratio, one paper again included siblings as a control group, and the other one had a small research group (n = 10), which could have led to the statistical irrelevance of the results. The studies with different results that are difficult to clarify are those by Strati et al. and Zhang et al. An explanation could not be obtained for the Firmicutes abundance trend. There was no clear trend of age-related differences either, despite a wide range of patient ages. This might stem from the fact that the microbiota of a child starts to resemble the adult one around age 3–5 years, which was the lowest age of patients included in the research groups of the analyzed studies [71,72].

The main question, of course, is whether we could use these observations for a further supportive therapy, targeted at the GI symptoms or affective and behavioral difficulties. Probiotics seem to be the easiest intervention, with some improvement observed after their introduction. Their exact mechanism of action is unclear; however, it may be based both on the simple dysbiosis improvement and the reduction of gut inflammation. On the gut–brain axis level, their influence on neurotransmission, based on serotonin and gamma-aminobutyric acid (GABA), as well as on the level of oxytocin, could lead to a behavioral and affective improvement [21]. The most promising probiotics included *Lactobacillus*, *Bifidobacterium*, and *Streptococcus* strains, which were effective for all the aforementioned aspects. They also led to an improvement in microbiota composition, especially by increasing the Bacteroidetes/Firmicutes ratio. From *Lactobacillus*, *L. acidophilus* seemed to provide a behavioral improvement both in LB and L only probiotics, but no improvement in the GI symptoms. MTT was a less researched method of treatment; however, it also led to improvements in FGID and behavior, which seemed to persist long after the intervention. An improvement in GI symptoms has been already observed in irritable bowel syndrome [73,74]. Moreover, the usage of MTT was studied in psychiatric disorders with promising results, including the reduction of anxiety and depression symptoms [75]. The greatest disadvantage of MTT is, however, its invasiveness, which can be discouraging, especially in children. Prebiotics seem to provide some improvement in GI and behavioral symptoms, both alone and combined with probiotics. However, the diverse probiotics used in the studies on ASD children make it difficult to form a clear conclusion. B-GOS was studied earlier in the adult group and led to the alleviation of FGID symptoms and anxiety [76,77], while bovine colostrum appears to be helpful in inflammatory bowel diseases and infectious diarrhea [78]. This suggests that, while promising, prebiotics should be used in a trial on a larger group of patients to assess their efficacy.

## 4. Conclusions

The emerging field of research on the relationship between the gut microbiota and autism spectrum disorder holds promise in shedding light on the intricate connections between the GI system and neurodevelopmental disorders. The complex interplay between the gut microbiome and various aspects of ASD, including the core symptoms, Gl symptoms, and behavioral difficulties, highlights the potential importance of the gut–brain axis in the etiology and manifestation of ASD.

Several trends emerge from the review’s findings. The altered composition of the gut microbiota in children with ASD and its correlation with GI symptoms and core behavioral characteristics warrants further investigation. Interventions, particularly probiotics and microbiota transfer therapy, show promise in ameliorating both the GI and behavioral symptoms. The positive outcomes reported in some studies suggest that targeting the gut microbiome could be a viable avenue to improve the quality of life of individuals with ASD. However, the specific mechanisms of action and the long-term effects of these interventions require more extensive exploration.

While this systematic review presents a comprehensive analysis of current research on the gut microbiota composition in children with ASD and the effectiveness of various interventions, it is limited by several factors. The microbiome’s diversity, its susceptibility to numerous factors, and its intricate connection to both physiological and neurological processes contribute to the nuanced findings reported across different studies. The variability in methodologies, study populations, and interventions, as well as small sample sizes and the lack of standardized protocols, hinder the ability to draw definitive conclusions or establish universal recommendations. Therefore, further large-scale, controlled studies are needed to validate the potential benefits of microbiota-based interventions for children with ASD.

In summary, this systematic review offers valuable insights into the intricate relationship between the gut microbiota and autism spectrum disorder. The findings highlight the potential influence of the gut–brain axis on the core symptoms, behavioral difficulties, and GI symptoms observed in ASD. While the results underscore the potential of microbiota-based interventions, the heterogeneity of the findings and the complexity of the gut microbiome emphasize the need for more rigorous and standardized research to unlock the full potential of this emerging field.

## Figures and Tables

**Figure 1 ijms-24-16660-f001:**
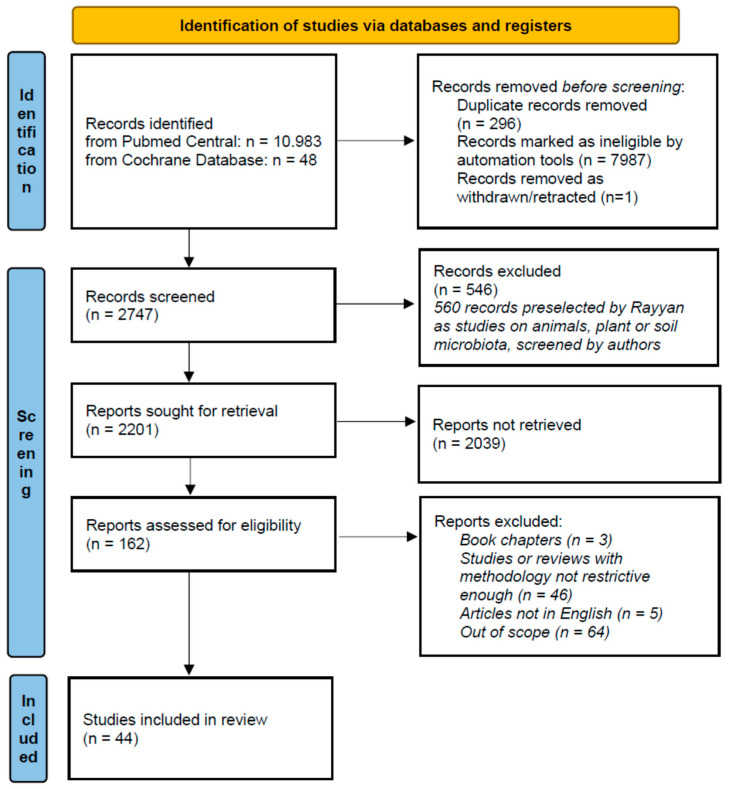
PRISMA 2020 flow diagram for systematic reviews.

**Figure 2 ijms-24-16660-f002:**
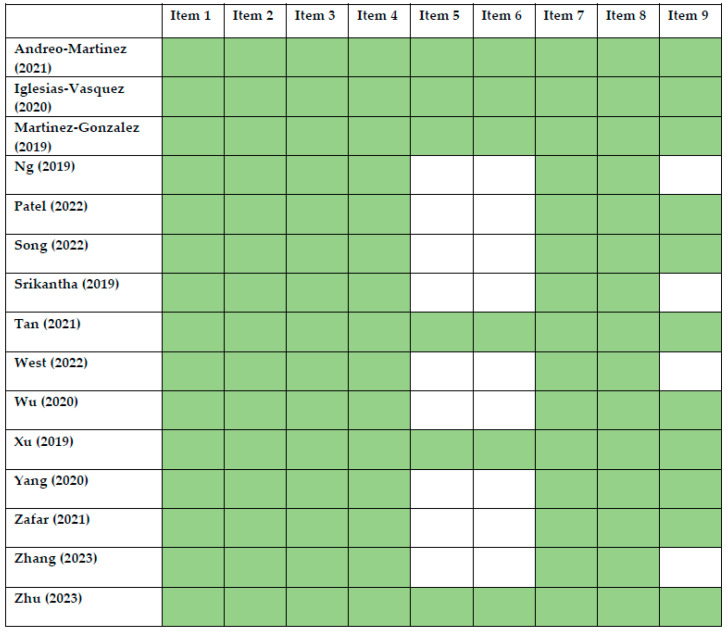
Quality assessment of systematic reviews and meta-analyses using AMSTAR Checklist. Item 1—PICO (population, intervention, control group, outcome) description; Item 2—Review methods established prior to review; Item 3—Explanation of selection; Item 4—Comprehensive literature search strategy; Item 5—Study selection in duplicate; Item 6—Data extraction from duplicate; Item 7—List of excluded studies or explanation; Item 8—Description of included studies; Item 9—Assessment of Risk of Bias [14,15,16,17,18,19,20,21,22,23,24,25,26,27,28].

**Figure 3 ijms-24-16660-f003:**
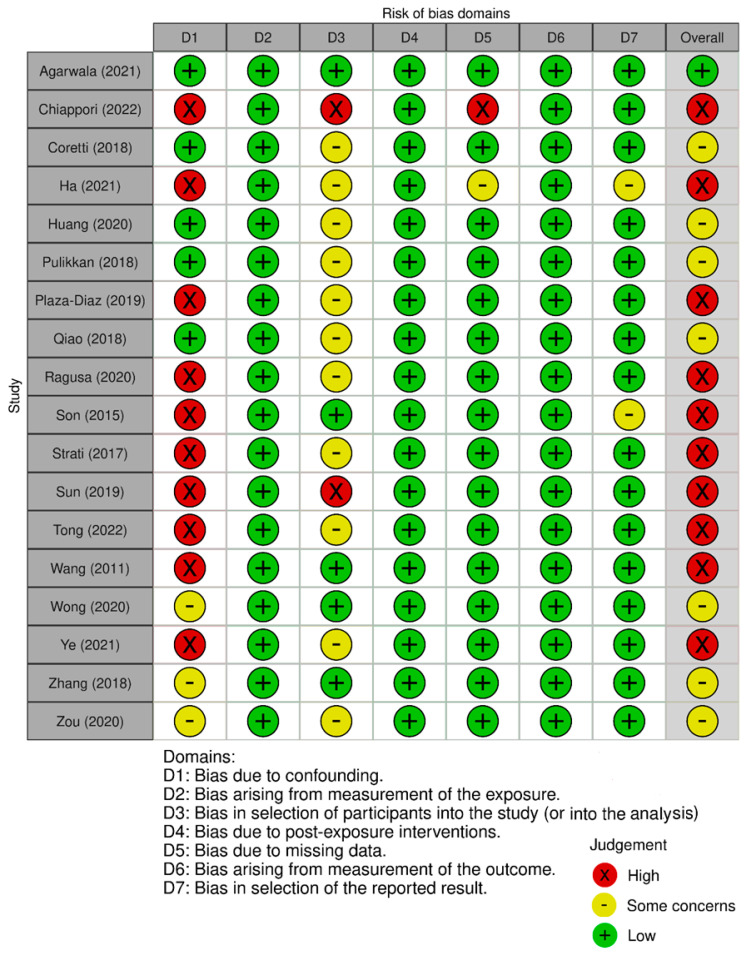
Non-RCTs without intervention—quality assessment using the Risk of Bias in Non-Randomized Studies of Exposure [29,30,31,32,33,37,38,39,40,42,43,44,47,48,49,50,51,52].

**Figure 4 ijms-24-16660-f004:**
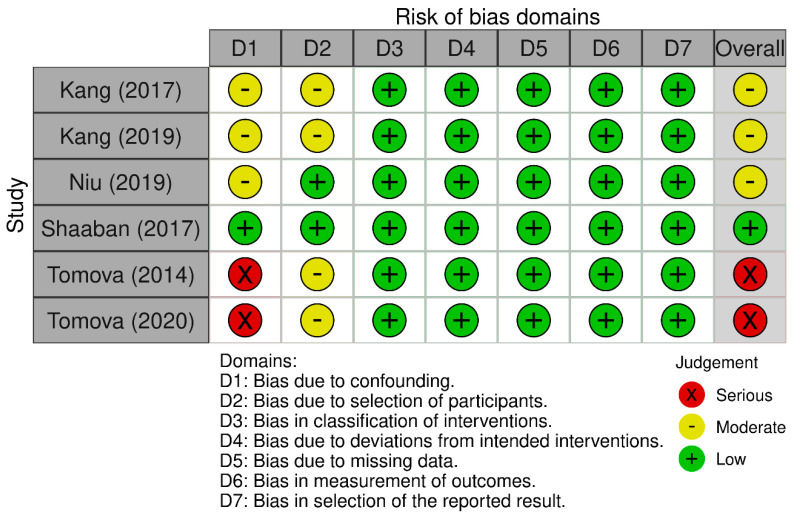
Non-RCTs with intervention—quality assessment using the Risk of Bias in Non-Randomized Studies of Interventions [34,35,36,41,45,46].

**Figure 5 ijms-24-16660-f005:**
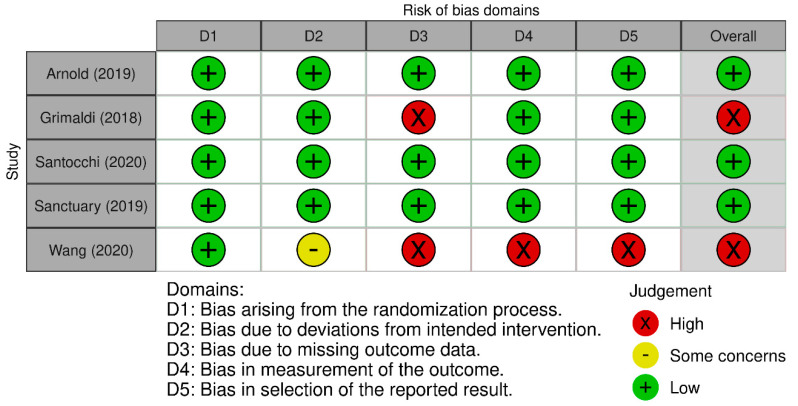
RCTs quality assessment using Cochrane revised tool for risk of bias in randomized trials [5,8,9,10,11].

**Table 1 ijms-24-16660-t001:** List of non-interventional original studies on gut microbiota included in the review.

Study	Country	ASD (n)	ASD Age (Years)	Healthy Control (n)	Healthy Control Age (Years)	Microbiota Assessment Method	Study Type	Microbiota Assessment —Other Important Results		
								Behavior	GI Symptoms	Microbiota Composition
Agarwala (2021) [29]	India	30	–	30	–	16s rRNA	Retrospective	–	–	–
Chiappori (2022) [30]	Italy	6	6–17	6	10–20	16s rRNA	case-controlled	–	–	–
Coretti (2018) [31]	Italy	11	2.92 ± 0.48	14	2.92 ± 0.70	16s rRNA	case-controlled	–	–	–
Ha (2021) [32]	South Korea	54	7.0 ± 2.1	38	6.0 ± 1.7	16s rRNA	case-controlled	weak correlation of microbiota composition with SRS	–	–
Huang (2020) [33]	China	39	4.74 ± 1.12	44 (healthy control) 38 (mother control)	5.11 ± 0.95 (healthy control) 34.12 ± 5.07 (mother control)	16s rRNA	case-controlled	–	–	–
Pulikkan (2018) [38]	India	30	3–16	24	3.5–16	16s rRNA	case-controlled	–	–	–
Plaza-Diaz (2019) [37]	Spain	54	2–6	57	2–6	16s rRNA	case-controlled	–	–	–
Son (2015) [42]	USA	66	10.3 ± 1.8	37 (siblings)	10.0 ± 1.8	16s rRNA	case-controlled	–	correlation of FGID with: - Firmicutes: *Asteroleplasma* - Proteobacteria: *Thalassospira*, *Burkholderia*, *Comamonadaceae* - Fusobacteria: *Fusobacteriales* - Bacteroidetes: *Prevotellaceae* - Actinobacteria: *Mobiluncus*	–
Strati (2017) [43]	Italy	40	5–17	40	3.6–12	16s rRNA	case-controlled	–	negative correlation of constipation with *Gemmiger*, *Ruminococcus* positive correlation of constipation with *Escherichia/Shigella*, *Clostridium cluster XVIII*	–
Sun (2019) [44]	China	9	3–12	6	3–12	16s rRNA	case-controlled	–	–	–
Tomova (2020) [46]	Slovakia	46	4.0–8.5	16	2.8–9.15	16s rRNA	case-controlled	–	–	–
Wang (2011) [48]	Australia	23	3–17	31 (22—siblings, 9—healthy independent)	siblings: 4.5–18.5 healthy independent: 3.5–15	qPCR	case-controlled	–	–	comparable levels of sulfate-reducing bacteria in ASD and neurotypical children
Wong (2020) [49]	China	92	8.43 ± 1.54	112	8.12 ± 1.99	16s rRNA	case-controlled	–	–	–
Ye (2021) [50]	China	71	4.28 ± 1.52	18	4.62 ± 1.29	16s rRNA	case-controlled	–	–	strains selected for ASD prediction: *Prevotella buccae*, *Bifidobacterium longum*, *Streptococcus thermophilus*, *Enterobacter cloacae*, *Klebsiella oxytoca*, *Eubacterium hallii*, *Clostridium ramosum*, *Erysipelotrichaceae bacterium 6_1_45*, *Eubacterium siraeum*, *Lautropia mirabilis*
Zhang (2018) [51]	China	35	4.9 ± 1.5	6	4.6 ± 1.1	16s rRNA	case-controlled	–	–	–
Zou (2020) [52]	China	48	5 (2–7)	48	4	16s rRNA	case-controlled	–	–	–

FGID—functional gastrointestinal disorder; SRS—the Social Responsiveness Scale.

**Table 2 ijms-24-16660-t002:** List of non-interventional original studies on oral microbiota included in the review.

Study	Country	ASD (n)	ASD Age (Years)	Healthy Control (n)	Healthy Control Age (Years)	Microbiota Assessment Method	Study Type	Microbiota Assessment—Behavior
Qiao (2018) [39]	China	32 (salivary) 26	10.02 ± 1.43 (salivary) 10.15 ± 1.35 (dental)	27 (salivary) 26 (dental)	10.19 ± 0.59 (salivary) 10.37 ± 0.66 (dental)	16s rRNA	case-controlled	–
Ragusa (2020) [40]	Italy	76	7 ± 1.5	39	6.75 ± 1.51	RT-PCR	case-controlled	positive correlation: *Moryella*—VIQ, *Ralstonia*—ADI-A negative correlation: *Moryella*—ADI-D, *Saccharibacteria*—ADI-B and ADI-C, *Weeksellaceae* and *Ralstonia*—VIQ, PIQ, TIQ significant predictors of *Tannerella* abundance: VIQ, PIQ, TIQ, ADI-C, ADOS-A
Tong (2022) [47]	China	26	4.13 ± 0.95	26	4.04 ± 0.89	16s rRNA	case-controlled	–

ADI—the Autism Diagnostic Interview; ADI-A—qualitative anomalies in social interactions; ADI-B—qualitative anomalies in communication; ADI-C—repetitive and restricted behavior; ADI-D—anomalies in neurodevelopment arising before 36 m.o.; ADOS—the Total Autism Diagnosis Observation Schedule; ADOS-A—communication; PIQ—the performance intelligence quotient; TIQ—the total intelligence quotient; VIQ—the verbal intelligence quotient.

**Table 3 ijms-24-16660-t003:** List of interventional original studies on gut microbiota included in the review.

Study	Country	ASD (n)	ASD Age (Years)	Healthy Control (n)	Healthy Control age (years)	Microbiota Assessment method	Study Type	Intervention Type	Intervention Length	Microbiota Assessment— Other Important Results		
										Behavior	GI Symptoms	Microbiota Composition
Arnold (2019) [5]	USA	10 (6—probiotic first, then placebo, 4—placebo first, then probiotic)	3–12	0	–	16S rRNA	RCT, placebo-controlled, double-blind, parallel	Probiotic (*Lactobacillus casei*, *L. plantarum*, *L. acidophilus*, *L. delbrueckii subsp. bulgaricus*, *Bifidobacterium longum*, *B. infantis*, *B. breve*, *Streptococcus salivarius ssp. thermophilus*)	8 weeks of probiotics or placebo, 3 weeks of washout, 8 weeks of a crossover treatment	statistically important improvement: ABC, SRS, CSHQ, PSI-SF non-statistically important improvement: PRAS-ASD	non-statistically important improvement: Peds-QL GI module correlation of relative abundance of *Lactobacillus* and PEDS-QL	–
Grimaldi (2018) [8]	UK	30 (18—non-restrictive diet, 12—exclusion diet)	4–11 (mean 7.7)	–	–	16s rRNA	RCT, double-blind, parallel	B-GOS	6 weeks	B-GOS: - some children showed improvement in sleep patterns - significant improvement in social behavior of children on restrictive diet	lower baseline FGID in the children on restrictive diet (abdominal pain, bowel movement) B-GOS: - non-statistically important improvement in FGID,	unrestricted diet group: increase in *Bifidobacterium* spp., *Ruminococcus* spp., *Lachnospiraceae* (*Coprococcus* spp., *Dorea formicigenerans*, *Oribacterium* spp.), *Eubacterium dolchum*, *Saccharibacteria*, *Mogibacteriaceae* restricted diet group: *Bifidobacterium adolescentis* and *Bifidobacterium longum* the most abundant in *Bifidobacterium* spp.—the latter predominant
Kang (2017) [34]	USA	18	7–16	20	7–16	16s rRNA	nRCT, cohort, case-controlled	MTT	2 weeks of antibiotic treatment and cleansing bowels, MTT for 7–8 weeks	significant improvement in behavior (PGI-II, CARS, VABS-II)	significant improvement in GI symptoms: abdominal pain, indigestion, diarrhea, constipation (GSRS)	increase in the abundance of *Bifidobacterium*, *Prevotella*, *Desulfovibrio* (two former—persistent increase)
Kang (2019) [35]	USA	18	7–16	–	–	16s rRNA	nRCT, cohort, case-controlled	MTT—follow-up after 2 years	2 weeks of antibiotic treatment and cleansing bowels, MTT for 7–8 weeks; follow-up after 2 years	–	–	–
Niu (2019) [36]	China	114; 37—probiotic, 28—without probiotic	3–8 (mean 4.5)	40	3–8 (mean 4.2)	PCR	nRCT, open-label, case-controlled	Probiotics	4 weeks	improvement in behavior (ATEC), mood, eating pattern abnormalities, sleep quality	improvement	–
Sanctuary (2019) [9]	USA	8	6.8 ± 2.4 (3.9–10.9)	–	–	16s rRNA	RCT, double-blind, placebo controlled, parallel	Prebiotic (bovine colostrum product) only or pre- and probiotic (bovine colostrum product + *Bifidobacterium infantis*)	5 weeks of probiotic + prebiotic supplementation, 2 weeks of washout, 5 weeks of prebiotic only supplementation	BCP only: - improvement in irritability, stereotypy, hyperactivity, lethargy—ABC, ABAS-II, RBS-R BCP + probiotic: - improvement in lethargy (ABC)	BCP only: - 87.5% of patients—some improvement in GI symptoms (QPGS-RIII, CHARGE-GIH) - GI symptoms improvement greater, according to parents BCP + probiotic: - 100% of patients—some improvement in GI symptoms (QPGS-RIII, CHARGE-GIH)	four microbiota enterotypes—high in: *Prevotella*, *Bifidobacterium*, *Bacteroides*, mixed no or inconsistent change in enterotype after intervention
Santocchi (2020) [10]	Italy	63	4.15 ± 1.08	–	–	–	RCT, double blind, parallel, factorial, efficacy controlled	Probiotic: *Streptococcus thermophilus*, *Bifidobacterium* *breve*, *B. longum*, *B. infantis*, *Lactobacillus acidophilus*, *L. plantarum*, *L.* *para-casei*, *L. delbrueckii subsp. bulgaricu*	6 months	no statistically significant difference in the total ADOS-CSS if analyzed in children both with and without GI symptoms in children without GI symptoms: the total ADOS-CSS decreased in probiotic group and increased in placebo group in children with GI symptoms: improvement in adaptive functioning (receptive, domestic, and coping skills, sensory profile—VABS-II subscales)	in children with GI symptoms: improvement (total GSI, 6-GSI, stool smell, flatulence)	–
Shaaban (2017) [41]	Egypt	30	5–9	30	5–9	RT-PCR, qPCR	nRCT, prospective, open-label	Probiotic (*Lactobacillus acidophilus*, *L. rhamnosus*, *Bifidobacterium longum*)	3 months	improvement in speech/language/communication, sociability, sensory/cognitive awareness, and health/physical/behavior in ATEC	improvement in 6-GSI (especially constipation, stool consistency, flatulence, abdominal pain)	increase in *Bifidobacterium* and *Lactobacillus*
Tomova (2014) [45]	Slovakia	10	2–9	19 (10—non-autistic siblings, 10—non-autistic independent controls)	siblings: 5–17 independent controls: 2–11	RT-PCR	nRCT, prospective, open-label, controlled	Probiotic: 3 strains of *Lactobacillus*, 2 strains of *Bifidobacteria*, 1 strain of *Streptococcus*	4 months	positive correlation between severity of core symptoms and higher *Clostridia* and *Desulfovibrio* levels, and lower Bacteroidetes/Firmicutes ratio	positive correlation between severity of GI symptoms and lower *Clostridia* and *Desulfovibrio* levels, and lower Bacteroidetes/Firmicutes ratio	siblings vs. independent controls: - ↑ Firmicutes - ↓ Bacteroidetes - ↓ Bacteroidetes/ Firmicutes ratio Probiotics: - increase in Bacteroidetes/Firmicutes ratio - lower *Lactobacillus* absolute amount, higher relative amount - lower *Desulfovibrio* level
Wang (2020) [11]	China	26	2–8	24	2–8	16s rRNA	RCT, case-controlled, placebo-controlled, double-blind	Probiotic (*Bifidobacterium infantis*, *B. lactis*, *Lactobacillus rhamnosus*, *L. paracasei*)	30 days	a significant decrease in severity of autistic symptoms—ATEC (speech/language/communication and sociability categories)	–	–

6-GSI—6-Item Gastrointestinal Severity Index; ABAS-II—the Adaptive Behavior Assessment System—Second Edition; ABC—the Aberrant Behavior Checklist; ADOS—the Total Autism Diagnosis Observation Schedule; ADOS-A—communication; ADOS-CSS—Calibrated Severity Score; ATEC—the Autism Treatment Evaluation Checklist; B-GOS—prebiotic galactooligosaccharide mixture; CSHQ—Tthe Children’s Sleep Habits Questionnaire; CHARGE-GIH—the Childhood Autism Risks from Genetics and Environment Gastrointestinal History Questionnaire; FGID—functional gastrointestinal disorder; GSRS—the Gastrointestinal Symptom Rating Scale; MTT—microbiota transfer therapy; nRCT—non-randomized controlled trial; Peds-QL GI Module—the Gastrointestinal Module of the Pediatric Quality of Life Inventory; PRAS-ASD—the Parent-Related Anxiety Scale for Autism Spectrum Disorder; PSI-SF—the Parenting Stress Index Short Form; QPGS-RIII—Pediatric Gastrointestinal Symptoms—Rome III Version; RBS-R—Repetitive Behavior Scale—Revised; RCT—randomized controlled trial; SRS—Social Responsiveness Scale; Total GSI—the Total Gastrointestinal Severity Index; VABS-II—the Vineland Adaptive Behavior Scale—Second Edition.

**Table 4 ijms-24-16660-t004:** List of meta-analyses and systematic reviews included in the review.

Study	Country	Articles Included (n)	ASD (n)	Healthy Control (n)	Method	Microbiota Assessment		
						Behavior	GI Symptoms	Microbiota Composition
Andreo-Martinez (2021) [14]	Spain	18	642	356	MA	*Bacteroides*—positive correlation with ASD severity	–	–
Iglesias-Vasquez (2020) [15]	Spain	18	493	404	SR/MA	–	–	–
Martinez-Gonzalez (2019) [20]	Spain	16	–	–	SR	–	–	–
Ng (2019) [21]	Singapore	8	544	–	SR	Probiotics: - *L. plantarum* (39 patients): no differences in behavior - *L. acidophilus* (22 patients): improvement in concentration - *L. acidophilus*, *L. rhamnosus*, *B. longum* (30 patients): improved behavior - *L. rhamnosus*, *B. animalis*, *B. lactis* (342 patients): worse behavior after probiotics - *L. acidophilus*, *L. casei*, *L. delbrueckii*, *B. longum*, *B. bifidum* (33 patients): improvement in behavior B-GOS (30 patients): - no difference in sleep pattern	Probiotics: - *L. acidophilus*, *L. casei*, *L. delbrueckii*, *B. longum*, *B. bifidum* (33 patients): improvement in GI symptoms - *L. plantarum* (39 patients): no differences in GI symptoms - *L. acidophilus*, *L. rhamnosus*, *B. longum* (30 patients): improved GI symptoms Bovine colostrum (8 patients): - ↓ GI symptoms B-GOS (30 patients): - ↓ GI discomfort with no difference in GI symptoms	Probiotics: - *L. plantarum* (39 patients): ↓ *Clostridium* after probiotics - *L. acidophilus*, *L. rhamnosus*, *B. longum* (30 patients): ↑ *Bifidobacterium* after probiotics - *Lactobacillus* (3 strains), *Bifidobacteria* (2 strains), *Streptococcus* (1 strain) (29) patients: *↓ Bifidobacterium*, *Lactobacillus*
Patel (2022) [22]	USA	9	710	–	SR	–	- gut dysbiosis connected with the severity and prevalence of GI symptoms - probiotics: improvement in GI symptoms and behavior - prebiotics: equivocal efficacy	–
Song (2022) [16]	China	3	144 (74 pro- and prebiotic, 34 placebo)		MA	- core symptoms: 2 studies—no significant improvement, 1 study—improvement in speech/language/communication - behavior: improvement in 2 studies	improvement in 2 studies	–
Srikantha (2019) [23]	Switzerland	136	–	–	SR	- ↓ diversity correlated with the severity of GI symptoms - ↑ *Clostridium* spp. (*C. perfingens*) correlated with higher severity of symptoms (CARS) Probiotics: - *Lactobacillus + Bifidobacterium:* improvement in PGI-III Vancomycin therapy: - improvement in behavioral difficulties during antibiotic therapy, but not after its end (weak evidence)	MTT: - improvement in GI symptoms Probiotics: improvement in GI symptoms	Probiotics: - *Lactobacillus + Bifidobacterium + Streptococcus:* ↑ Bacteroidetes and Bacteroidetes/Firmicutes ratio - *Lactobacillus + Bifidobacterium:* -↑ diversity - ↑ *Bifidobacterium*, *Prevotella*, *Desulfovibrio*
Tan (2021) [24]	Canada	13	481	–	SR	Probiotics: - no influence, 4 RCTs - positive influence, 5 papers (1 RCT, 4 non-RCTs) Pre- and synbiotics: - positive influence, 4 papers (3 RCTs, 1 non-RCT) MTT: - positive influence, 1 non-RCT study in 2 papers	Probiotics: - no influence, 4 papers (3 RCTs, 1 non-RCT) - positive influence, 2 non-RCTs Pre- and synbiotics: - no influence, 1 RCT - positive influence, 2 papers (1 RCT, 1 non-RCT) MTT: - positive influence, 1 non-RCT study in 2 papers	–
West (2022) [17]	USA	13	–	–	MA	–	–	–
Wu (2020) [25]	China	5	169	128	MA	–	–	*Prevotella*, *Roseburia*, *Ruminococcus*, *Megasphaera*, *Catenibacterium*—potential biomarkers of ASD (in forest analysis machine learning)
Xu (2019) [18]	China	9	254	167	SR, MA	–	–	–
Zafar (2021) [27]	Pakistan	8 (6 original papers + 2 SRs including 153 studies)	330	178	SR	–	–	–
Zhang (2023) [28]	China	5	150	–	SR	MTT: - improvement in CARS, ABC, VABS-II, sleep disturbances	MTT: - improvement - ↓ *Eubacterium coprostanoligenes* correlated with GI symptoms	MTT: - *↑ Bifidobacterium*, *Prevotella*, *Desulfovibrio*, *Roseburia*, *Ruminococcus*, *Faecalibacterium*
Zhou (2023) [52]	China	9 (5 RCTs included in MA)	186	150	MA	MTT: ABC and CARS significantly ↓	–	–
Yang (2020) [26]	China	16	376	66	SR	Probiotics: - improvement in ASD core symptoms comparable to placebo, 3 papers - greater improvement after probiotics only in comparison to probiotics + prebiotics, 1 paper - improvement in irritability (ATEC), 2 papers - improvement in concentration and carrying out orders, 1 paper Prebiotics: - improvement in sociability, 1 paper Pre- and probiotics combined: - improvement in behavior (CARS, SRS, VABS-II, ABC), 2 papers MTT: - ↓ CARS, 1 paper Vitamin A supplementation: - no influence on ASD core symptoms *Streptococcus*, *Alistipes putredinis*, *Bacteroides*, *Clostridium*—positive correlation with ASD symptoms *Coprococcus*—negative correlation with CARS *Lactobacillus*—correlation with Peds-QL	Probiotics: - no improvement, 2 papers - non-statistically important improvement, 2 papers - statistically important improvement, 1 paper Prebiotics: - improvement, 1 paper Pre- and probiotics combined: - improvement, 2 papers MTT: - statistically significant improvement, 2 papers *Blautia wexlerae*—positive correlation with the frequency of stool passage *Clostridium leptum*, *Eubacterium sp. Marseille*—negative correlation with the frequency of stool passage	Probiotics: - effective, 8 papers - ineffective, 1 paper - ↑ *Bifidobacteria*, *Lactobacillus* (relative amount), *Enterococci* - ↓ *Firmicutes*, *Desulfovibrio*, *Clostridium*, *Lactobacillus* (absolute amount) Prebiotics: - ↓ a-diversity - ↑ *Bifidobacteriales* - ↓ *Clostridium* MTT: - ↑ *Bifidobacterium*, *Prevotella*, *Desulfovibrio* - ↓ *Bacteroides fragilis* Vitamin A supplementation: - ↑ Bacteroidetes/Bacteroidales and Bacteroidetes/Firmicutes ratio

ABC—the Aberrant Behavior Checklist; ASD—autism spectrum disorder; ATEC—Autism Treatment Evaluation Checklist; B-GOS—prebiotic galactooligosaccharide mixture; CAR—Childhood Autism Rating Score; GI symptoms—gastrointestinal symptoms; MA—Meta-analysis; MTT—microbiota transfer therapy; Peds-QL GI Module—The Gastrointestinal Module of the Pediatric Quality of Life Inventory; PGI-III—Parental Global Impressions III; SR—systematic review; SRS—Social Responsiveness Scale; VABS-II—The Vineland Adaptive Behavior Scale—Second Edition; ↑—increase; ↓—decrease.

**Table 5 ijms-24-16660-t005:** Microbiota in ASD patients vs. control—gut microbiota.

Phylum/*Genus*	Agarwala (2021) [29]	Arnold (2019) [5]	Chiappori (2022) [30]	Coretti (2018) [31]	Grimaldi (2018) [8]	Ha (2021) [32]	Huang (2020) [33]	Kang (2017) [34]	Niu (2019) [36]	Pulikkan (2018) [38]	Plaza-Diaz (2019) [37]	Sanctuary (2019) [9]	Shaaban (2017) [41]	Son (2015) [42]	Strati (2017) [43]	Sun (2019) [44]	Tomova (2014) [45]	Tomova (2020) [46]	Wang (2011) [48]	Wang (2020) [11]	Wong (2020) [49]	Ye (2021) [50]	Zhang (2018) [51]	Zou (2020) [52]
** *Actinobacteria* **			↓			↑	↑		↓		↑									↓	↓ng	↑		↑
*Actinomyces*				↓																				
*Bifidobacterium*	↓		↓	↓	↓ed ↑nd	↑		↓	↓	↑		↓	↓				↑ sc		↓sc, ic	↓		↓		↑
*Collinsella*	=						↑								↑						↓ng			
*Corynebacterium*	=			↓																				
*Eggerthella*			↓	↓	↑nd																			
*Nitriliruptor*																		↑						
** *Bacteroidetes* **			↓		↑ed	↓			↓					=		↓				↑		↓		↑
*Alistipes*			↓												↓						↓ng			
*Barnesiella*			↓																					
*Odoribacter*																				↑				
*Parabacteroides*	=		↓	↑			↓								↓			↑nfs			↓ng			
*Prevotella*	↑		↑					↓		↓				=				↑fs	=					↑
** *Bacillota * ** ** *(Firmicutes)* **			↓		↑ed	=	↑3,4 ↓5,6		↑	↑	↑3,7,8 ↓6					↓9 ↑5					↑g	↑/ ↓5		↓/↑5,6
*Acidaminococcus*			↑																					
*Anaerophilum*																		↑fs						
*Anaerostipes*																								↑
*Blautia*	=								↓												↑ng	↑		↑
*Butyricicoccus*							↓																	
*Butyrivirio*										↑														
*Christensenella*			↓																					
*Cloacibacillus*			↑																					
*Clostridium*			↓10	↑													↑sc	↑fs	=		↓ng			↑
*Coprococcus*										↑														
*Dehalobacterium*					↑nd																			
*Dialister*	↓		↓												↓						↑g	↓		↓
*Dorea*															↑						↑ng			
*Eisenbergiella*																						↑		
*Enterococcus*	=										↑								=					
*Eubacterium*																						↓		↑
*Faecalibacterium*	↓		↓	↑	↑ed					↓									=			↑	↓	↑
*Filifactor*																								
*Flavonifactor*																								↓
*Fusicatenibacter*																					↑g			
*Gemella*				↓																				
*Lactobacillus*	=						↑			↑					↑			↑						
*Lactostreptococcus*					↑nd																			
*Lachnospira*	↓		↓						↑															↑
*Limosilactobacillus*			↑																					
*Megasphaera*										↑												↓		
*Mitsuokella*										↑														
*Oscillospira*				↑																↑				
*Phascolarctobacterium*	=		↓																		↓g			
*Streptococcus*	↓		↓	↓	↑nd																	↓	↓	↑
*Roseburia*	=				↑ed		↓		↓	↓													↓	↑
*Ruminococcus*	=		↑	↑					↓	↑										↑				
*Ruminiclostridium*																						↓		
*Sarcina*	↑																							
*Turicibacter*																					↑g			
*Veillonella*	↑														↓							↓	↓	
*Fusobacteriota*						=					↑													
*Cetobacterium*																				↑				
*Fusobacterium*	↑																							
**Lentisphaerota**																				↑				
** *Proteobacteria (Pseudomonadota)* **			↑	↑		=	↑		↑		↑							↑fs				↑	↓	
*Citrobacter*	↓						↑															↓		
*Constrictibacter*																		↑						
*Dichelobacter*																		↑						
*Diaphorobacter*																		↓						
*Enterobacter*							↑																	
*Escherichia*																			=			↓	↓	↓
*Haemophilus*	=																							↓
*Klebsiella*										↑												↑		
*Nitratireductor*																		↓						
*Phyllobacterium*							↑																	
*Providencia*																						↓		
*Salmonella*																		↑fs						
*Shigella*																					↓ng	↓		
*Sutterella*	↑		↑																					
** *Saccharibacteria* **														=										
** *Thermodesulfobacteriota* **										↑	↑													
*Bilophila*															↓									
*Desulfovibrio*								=									↑sc							
** *Verrucomicrobiota* **						=																		
*Akkermansia*	↓		↓		↑nd														↓ic			↓	↓	↓
** *Bacteroidetes/* ** ***Firmicutes* ratio**				↑						↓					↑	↓				=	↓	↓	↑	↓
**Overall bacterial diversity**		=		↑																↓	↓			
** *Escherichia/* ** ***Shigella* ratio**							↑											↑fs						

b—microbiota in the cecum and the terminal ileum biopsies; d—dental microbiota; ed—elimination diet; fs—food selectivity; g—gastrointestinal symptoms; ic—in comparison to independent control; nd—non-elimination diet; ng—no gastrointestinal symptoms; s—salivary microbiota; sc—in comparison to sibling control; ↑—higher abundance in ASD patients; (↑)—slightly (non-statistically significant) higher abundance in ASD patients; ↓—lower abundance in ASD patients; (↓)—slightly (non-statistically significant) lower abundance in ASD patients; =—similar abundance in ASD and neurotypical patients; 3—*Erysipelotrichaceae*; 4—Clostridiales family XIII; 5—*Ruminococcaceae*; 6—*Lachnospiraceae*; 7—*Bacillaceae;* 8—*Enterococcaceae*; 9—*Selenomonadales;* 10—CAG-352.

**Table 6 ijms-24-16660-t006:** Microbiota in ASD patients vs. control—meta-analyses and systematic reviews.

Phylum/*Genus*	Andreo-Martinez (2021) [14]	Iglesias-Vasquez (2020) [15]	Martinez-Gonzales (2019) [20]	Srikantha (2019) [23]	West (2022) [17]	Xu (2019) [18]	Zafar (2021) [27]
** *Actinobacteria* **							↑ (2)
*Actinomyces*				↑b			
*Bifidobacterium*	↓	↓		↓		↓	↓ (2)
*Collinsella*				↓			↑ (1)
** *Bacteroidetes* **		↑					↓ (2)/↑ (1)
*Alistipes*							↓ (1)
*Bacteroides*	(↑)	↑		↓b	↓ng	↓	
*Barnesiella*				↑			
*Odoribacter*				↑			
*Parabacteroides*		↑		↑		↓	↑ (1)/↓ (1)
*Porphyromonas*				↑			
*Prevotella*	(↓)		↓g	↑/↓b			↑ (2)/↓ (3)
** *Bacillota * ** ** *(Firmicutes)* **		↑					↑ (1)
*Acidaminococcus*							↓ (1)
*Anaerophilum*				↑			
*Clostridium*	(↑)	↑		↑	↓ng	=	↑ (3)
*Coprococcus*		↓					
*Dialister*							↓ (1)
*Dorea*				↑			↑ (1)
*Enterococcus*				↓		↓	
*Faecalibacterium*		↑		↑/↑b		↑	
*Flavonifactor*							↓ (1)
*Granulicatella*					↓		
*Lactobacillus*	(↑)			=		↑	↑ (1)/↓ (2)
*Lachnospira*	(↓)			↑b			↑ (1)/↓ (1)
*Lactococcus*				↓			
*Masilloclostridium*					↓ng		
*Oscillospira*				↓/↑b			↑ (1)
*Peptostreptococcus*				↑b			↑
*Phascolarctobacterium*		↑					
*Sporobacter*				↓			
*Staphylococcus*				↓			
*Streptococcus*	↓		↓g	↓/↓b			↑ (2)/↓ (1)
*Subdoligranulum*				↓			
*Roseburia*	(↓)			↑			
*Ruminococcus*	(↓)			↑b		↑	↑ (1)/↓ (1)
*Turicibacter*				↑			
*Tyzzerella*							↓
*Veillonella*			↓g				↑ (2)
** *Fusobacteriota* **							
*Fusobacterium*				↓			↓ (1)
** *Lentisphaerota* **							
** *Proteobacteria * ** ** *(Pseudomonadota)* **		↑		↑b			↑ (3)
*Aeromonas*				↑			
*Burkholderia*				↑b			
*Devosia*				↓b			↓
*Neisseria*				↓b			
*Parasutterella*				↑			
*Pseudomonas*				↑			
*Ralstonia*				↑b			
** *Tenericutes* **		↑					
*Enterobacter*				↑			↓ (1)
*Escherichia*				↓/↓b		↓	↑ (1)
*Shigella*							↓ (1)
*Sutterella*	(↑)			↑b			↑ (2)
** *Thermodesulfobacteriota* **							
*Bilophila*							↓ (1)
*Desulfovibrio*				↑			
** *Verrucomicrobiota* **				↓			↓ (1)
*Akkermansia*				↑		↓	↓ (1)
** *Bacteroidetes/* ** ***Firmicutes* ratio**				↓		↑	

b—microbiota in the cecum and the terminal ileum biopsies; g—gastrointestinal symptoms; ng—no gastrointestinal symptoms; ↑—higher abundance in ASD patients; (↑)—slightly (non-statistically significant) higher abundance in ASD patients; ↓—lower abundance in ASD patients; (↓)—slightly (non-statistically significant) lower abundance in ASD patients; (number)—number of papers including presented result.

**Table 7 ijms-24-16660-t007:** Microbiota in ASD patients vs. control—oral microbiota.

Phylum/*Genus*	Qiao (2018) [39]	Ragusa (2020) [40]	Tong (2022) [47]
** *Actinobacteria* **	↓	↑	
*Actinomyces*	↓s		
*Rothia*	↑d/↓s	↑	
*Bacteroidetes*	↓	↑	
*Alloprevotella*	↓ s, d		
*Porphyromonas*	↓ s		
*Prevotella*	↓ d/= s		
*Tannerella*		↓	
** *Bacillota (Firmicutes)* **	↓	↓	
*Filifactor*		↑	
*Moryella*		↓	
*Peptostreptococcus*	↓ s, d		
*Selenomonas*	↓ d		
*Solobacterium*	↓ s, d		
*Streptococcus*	↑ d		↑
** *Fusobacteriota* **	↓	↓	
*Fusobacterium*	↓ s, d		
*Leptotrichia*	↓ s, d		
** *Proteobacteria (Pseudomonadota)* **	↑	↑	
*Actinobacillus*		↑	
*Aggregatibacter*		↑	
*Haemophilus*	↑ s	↑	
*Ralstonia*		↑	
*Saccharibacteria*		↓	
**Overall bacterial diversity**	↓		

b—microbiota in the cecum and the terminal ileum biopsies; d—dental microbiota; ed—elimination diet; fs—food selectivity; g—gastrointestinal symptoms; ic—in comparison to independent control; nd—non-elimination diet; ng—no gastrointestinal symptoms; s—salivary microbiota; sc—in comparison to sibling control; ↑—higher abundance in ASD patients; (↑)—slightly (non-statistically significant) higher abundance in ASD patients; ↓—lower abundance in ASD patients; (↓)—slightly (non-statistically significant) lower abundance in ASD patients; =—similar abundance in ASD and neurotypical patients; (number)—number of papers including presented result.

## Data Availability

Not applicable.

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
