# Peer review of "Microbiota in Autism Spectrum Disorder: A Systematic Review"

_ijms, 2023, doi:10.3390/ijms242316660_

Round 1
Reviewer 1 Report
Comments and Suggestions for Authors
1. Does alteration in microbiota correlate with the severity of ASD in patients?
2. In the cited studies, there was a wide range of patient age. I was wondering if the age will affect the microbiota composition and if the ASD-related microbiota alteration has any age specificity.
3. Also, the patients from the analyzed studies came from very diverse background (from different countries). I was wondering if their regular dietary habit has any impact on the microbiota composition and leads to the discrepancy in the changes detected in those studies.
Comments on the Quality of English LanguageOverall, this manuscript is well-written.
Author Response
Dear Reviewer,
We would like to express our sincere gratitude for your invaluable contribution as a reviewer for our scientific paper titled Microbiota in Autism Spectrum Disorder: A systematic review”. Your expertise and thorough examination significantly enhanced the quality of our publication. In answer to your questions:
- Yes, we can observe such correlation – we describe it briefly in lines 138-142 and 398-400. Apart from the information in text, we include the information also in the table: Table 3 (Shaaban, Wang) and Table 4 (Andreo-Martinez, Ng, Song).
- We did not observe any clear age-specific trends in microbiota composition – it might relate to the fact, that the gut microbiota resembles the adult one around age of 3 to 5 years old (depending on the source). We included this information in lines 420-424.
- A typical trend for “Westernized” diet was a higher prevalence of Prevotella, mentioned in the discussion (lines 378-382). Apart from that, we observed no clear trend in comparison of results between the Western diet vs the non-Western diet (we included information about it lines 400-402).
Once again, thank you for your time, expertise, and valuable input. We look forward to the opportunity to collaborate with you again in the future.
Best Regards,
Authors.

Reviewer 2 Report
Comments and Suggestions for Authors
In this systematic study, the authors reviewed the gut microbiota in children with ASD and interventions targeting microbiota modulation. Indeed, ASD, as stated by the authors, is a complex neurodevelopmental condition characterized by core symptoms: restricted interests, communication difficulties, and impaired social interactions. Many ASD children may experience gastrointestinal functional disorders, impacting their well-being. Emerging evidence suggests a gut microbiota imbalance may exacerbate core and gastrointestinal symptoms.
The authors reviewed 44 studies (meta-analyses, reviews, and original research) to reveal insights into the gut microbiota-ASD relationship. They showed that ASD children exhibit increased Firmicutes and Pseudomonadota, and decreased Bacteroidetes. The Bacteroidetes to Firmicutes ratio tends to be lower, which correlates with behavioral and gastrointestinal abnormalities.
The authors also explained that probiotics, particularly Lactobacillus, Bifidobacterium, and Streptococcus strains, show promise in alleviating behavioral and gastrointestinal symptoms. Microbiota Transfer Therapy (MTT) seemed to have lasting benefits on microbiota and symptoms. Prebiotics potentially help with gastrointestinal and behavioral issues, need further research for conclusive efficacy.
The paper is well-written and belongs to the category of dozens and dozens of reviews dealing with ADS and microbiota.
Indeed, this is the main limit of this paper. The absence of novelty. This issue should be addressed by the authors.
Minor points:
-The name of the genus and of the species should be written in italics, also in the abstract.
-The English is easy to follow. However, the editing of the tables and pictures could be improved.
-In Figure 1 the authors retrieved 2201 papers but they excluded 2039. This is quite weird. Why?
-The authors included in the review 44 papers, according to the inclusion/exclusion criteria, with meta-analyses, reviews, and original research papers. This is also weird. Combining different kinds of works might elevate the bias. Why did the authors not restrict the work to original research papers only?
Minor editing of the English language is required
Author Response
Dear Reviewer,
We would like to express our sincere gratitude for your invaluable contribution as a reviewer for our scientific paper titled Microbiota in Autism Spectrum Disorder: A systematic review”. Your expertise and thorough examination significantly enhanced the quality of our publication. We want to answer to your suggestions.
We know the microbiota composition in ASD patients is a field of a particular interest of researchers in last few years, what leads to publishing multiple papers, both original ones and reviews. However, in 2023, according to our best knowledge, only one systematic review was published (Korteniemi et al. in Acta Psychiatrica Scandinavica). This systematic review did not present the details of differences in microbiota composition and possible interventions. The authors did not include information about the influence of microbiota abnormalities on behavioral symptoms. What is more, the beforementioned review did not use any tools for quality assessment of included studies, what was performed by us.
We corrected the italics in the genus and the species names. We also edited the first tables (Table 1-2 before, now renamed as Table 1-6) to make it clearer for the reader to follow the results.
The excluded articles (2039) concerned mainly the studies on other diseases which were listed by PubMed search engine because some of the words from query was mentioned briefly in article (for example, as a form of comparison between ASD and other neuropsychiatric disorders). Those papers were not retrieved. The process was performed using Rayyan Tool.
We included both original papers, meta-analyses, and systematic reviews, because we aimed at analyzing the data as thoroughly as possible. We decided to compare the results from single reports with what the meta-analyses and systematic reviews presented. However, we did not generalize the observation from both types of analyzed papers. We believe that presenting the results in this way we can give the readers complex summary of what the current state of knowledge is. However, the analysis of non-original reports may be easily excluded from the manuscript if it is necessary.
Once again, thank you for your time, expertise, and valuable input. We look forward to the opportunity to collaborate with you again in the future.
Best Regards,
Authors.

Reviewer 3 Report
Comments and Suggestions for Authors
The authors aim to review the evidence for a link between ASD and gut microbiota composition, and to describe the efficacy of interventions.
The topic is of interest to a wide range of investigators and health care professionals.
Comments:
The writing style should follow proper paragraph design, with a topic sentence and body and a transition to the next paragraph's topic. Many of the paragraphs in the manuscript contain only one or two sentences. This leads to a choppy manuscript with a lack of flow.
The abstract would be better if more detail was provided such as stating the % of articles reporting a specific outcome (e.g., "A majority (79%) of the reports showed reduced GI symptoms in response to daily probiotic consumption..."). It is of limited value to readers to state that an intervention shows promise.
Sentences that need English editing:
lines 48-51: you state that microbiota disrupt brain development... This is not correct and the sentence needs restructuring/modification.
line 53: perhaps specify "changes in the microbiota..."
line 55-56: BECAUSE the studies...remain equivocal.
line 60: avoid the term "proven" since single studies do not prove something, they can only support the observation of an effect.
Results
All of the statements of findings need to be referenced. Example: line 140: "In two of the meta-analyses (cite the papers)"
line 79: what does "advantage" mean?
It would be much more helpful to report the findings of changes in taxons as % of studies reporting the change vs. those showing no change.
Separate with subheadings the papers studying microbiota characterization 1) and GI symptoms vs. 2) without GI symptoms 3) ASD behaviors 4) without ASD behaviors 5) both GI and ASD.
Can you describe the length of the pro- and pre-biotic interventions in relation to efficacy?
Suggest separating figures into 1, 2, 3 rather than 1a, 1b...
Table 1: what were the microbiota outcomes for the studies showing none, such as Agarwala?
Table 3c: indicate the meaning of the numbers in the column for Zafar.
Reference 679: needs correction
Comments on the Quality of English LanguageConsiderable English editing is needed for proper sentence and paragraph structure.
Author Response
Dear Reviewer,
We would like to express our sincere gratitude for your invaluable contribution as a reviewer for our scientific paper titled Microbiota in Autism Spectrum Disorder: A systematic review”. Your expertise and thorough examination significantly enhanced the quality of our publication. We want to answer to your suggestions.
We improved the paragraph division in the manuscript. We also included the detailed data (percentage) in the abstract and in the results. All the results are referenced, and the reference previously in line 679 was corrected. We also corrected lines 48-51 (now: 50-51), 53 (now: 53-55), 55-56 (now: 56-58), 60 (now: 61), 79 (now: 84-85).
We separated tables into Tables 1-7. The results in tables are separated into those concerning GI symptoms and behavioral symptoms, as well as microbiota composition itself. In Table 1 – some authors, such as Agarwala, reported data that could be included in Tables 5-7; for the others, some important conclusions did not suit Tables 5-7 and were included in the Table 1 as “other important results” in microbiota assessment. In Table 3c (now: Table 7) the numbers are described under the table – for clarity, we added a more precise description. The length of the intervention was also included in Table 3.
Once again, thank you for your time, expertise, and valuable input. We look forward to the opportunity to collaborate with you again in the future.
Best Regards,
Authors.

Round 2
Reviewer 2 Report
Comments and Suggestions for Authors
The tables are hard to follow. The authors should consider the support of a professional in editing
Author Response
Dear Reviewer 2,
I hope this message finds you well. I wanted to express my sincere gratitude for your thoughtful and thorough review of our manuscript. Your insights have been instrumental in improving the quality of our work.
I am pleased to inform you that, in accordance with your valuable suggestions, we have made several revisions to the manuscript with the assistance of a computer graphics specialist. The scope of the introduced changes is outlined below:
- Minimized text wrapping in tables for improved readability.
- Aligned column positions in tables, particularly for Table 5, and merged cells where necessary.
- Changed "ned – non-elimination diet" to "nd" in Table 5 and beyond to accommodate more text in the column.
- Increased table caption font size to 9pt per MDPI requirements.
- Adjusted table text font size to 8pt per requirements.
- Increased arrow font size to 10pt for better visibility.
- Centered table content as per requirements.
- Enhanced table appearance, including edge style, color, text style, and text alignment.
- Added repeated table captions in the form of "Table X. Cont." for multi-page tables, as per guidelines.
- Repeated headers on each page for multi-page tables to enhance readability.
- Removed dark blue text color inside Table 5 and beyond to comply with journal instructions.
- Eliminated italic styling for numbers in Table 5 and beyond, aligning with footer indications.
- Replaced dashes with hyphens where necessary.
- Changed "N/A" and empty cells to hyphens as suggested by the journal (except for tables with "=" signs to maintain clarity).
- Adjusted fonts accordingly, retaining serif fonts in the text for better readability and using sans-serif fonts in tables for a visually coherent layout.
- Removed the unused "numer" symbol in Table 5.
- Completed the footer for Table 7.
- Added additional clarification to Table 5 regarding the use of arrow symbols.
Your guidance has significantly contributed to refining our manuscript. We sincerely appreciate your time and expertise in this process.
Best regards,
Authors
Reviewer 3 Report
Comments and Suggestions for Authors
The authors have made the revisions requested and the manuscript is improved in organization.
One comment:
Line 16: "...compared to children without ASD." "..which correlates..."
Comments on the Quality of English LanguageMinor edits are needed.
Author Response
Dear Reviewer 3,
I wanted to express my sincere gratitude for your thorough review of our manuscript. Your insights have been immensely valuable.
I would also like to inform you that we have addressed your comments, and in particular, we have supplemented information on line 16 of the manuscript.
Thank you once again for your time and expertise.
Best regards,